# Multi-stakeholder perspectives on reproductive and adolescent healthcare schemes in tribal regions of India: A qualitative study

Rohit Raj[1], Jarina Begum[1]*, Syed Irfan Ali[1‡], Manasee Panda[2‡]

1 Department of Community Medicine, Manipal Tata Medical College, Manipal Academy of Higher Education, Manipal, India, 2 Department of Community Medicine, Bhima Bhoi Medical College, Balangir, Odisha, India

◉ These authors contributed equally to this work.
‡ These authors also contributed equally to this work.
* jarina.begum@manipal.edu, dr.jarina@gmail.com

## Abstract

Tribal populations, comprising 8.9% of India's population, remain highly vulnerable to poor health outcomes due to socioeconomic deprivation, cultural barriers, and limited access to healthcare. Tribal women are disproportionately affected, facing high maternal mortality, poor reproductive health, and inadequate nutrition. Although government schemes such as Janani Suraksha Yojana (JSY) and Rashtriya Kishori Swasthya Karyakram (RKSK) aim to improve outcomes, utilization remains low. This study explored challenges and enablers in reproductive healthcare provision from multiple stakeholder perspectives in tribal Jharkhand. A descriptive qualitative study was conducted from December 2024 to June 2025 in two blocks of East Singhbhum district, Jharkhand. Using purposive sampling, 38 in-depth interviews were undertaken with stakeholders including tribal women, adolescent girls, Accredited Social Health Activists (ASHAs), Auxiliary Nurse Midwives (ANMs), Anganwadi Workers (AWWs), medical officers, and village heads. Interviews were conducted in Hindi, audio-recorded, translated, and transcribed into English. Thematic analysis was performed using Atlas.ti (version 8). Key challenges reported from the framed nine categories. Absenteeism of healthcare workers, inadequate infrastructure, cultural reliance on traditional medicine, weak governance, delayed incentive payments, financial and transport barriers, and limited awareness and use of health technology were the important themes developed. Enabling themes included supervision by higher authorities, quality service delivery, support from community leaders, and collaborations with NGOs. Significant barriers persist in delivering reproductive health schemes in tribal Jharkhand. Strengthening governance, ensuring workforce accountability, improving infrastructure, timely incentives, and implementing culturally tailored awareness interventions are essential to enhance reproductive health outcomes among tribal women and adolescents. Trial Registration: This study is

**Data availability statement:** All relevant data are within the manuscript and its Supporting Information files.

**Funding:** The author(s) received no specific funding for this work.

**Competing interests:** The authors have declared that no competing interests exist.

registered at Clinical Trial Registry-India (CTRI) with CTRI No: CTRI/2024/03/063824 on March 8, 2024.

## Introduction

The Tribal population of India constitutes roughly 8.9 percent of the total population in the country, is considered to be one of the most vulnerable populations concerning health [1]. The socioeconomic deprivation prevalent among tribal populations, limits the communities' access to essential resources that could include education and healthcare services makes these challenges even more challenging [2]. The reason that healthcare schemes need to be directed towards women in the tribal areas is because of the proportionate high maternal mortality rate, poor reproductive health, and inadequate nutritional status that characterizes these communities. Tribal women suffer more from all these issues because of cultural limitations placed upon them, which prevent them from moving out of the house freely and taking their own decisions within the family. It is, therefore, crucial to provide healthcare schemes to reverse the multiple layers of disadvantage that tribal women face. However, despite numerous government schemes that have been launched to enhance maternal and reproductive health outputs-for example, the Janani Suraksha Yojana (JSY), [3] and Rashtriya Kishori Swasthya Karyakram (RKSK), [4] there is disparity in maternal healthcare usage in the tribal region of Jharkhand as per NFHS 5 report [5].

While several studies in India have examined maternal and adolescent health among tribal populations, most have focused on service utilization, barriers, access, beneficiary awareness, or outcomes related to individual schemes in isolation [6,7]. There remains a critical gap in understanding how reproductive and adolescent health schemes function at the implementation level within tribal settings when viewed through the combined perspectives of beneficiaries, frontline health workers, health system managers, and local governance actors [6]. In particular, limited qualitative evidence exists that simultaneously captures health system, governance, cultural, and workforce-related barriers across multiple reproductive health schemes within a single tribal context.

In addition to policy implementation, the success or failure of health care schemes in tribal regions ultimately depends on the perceptions of various stakeholders concerned. The stakeholders in tribal health include: beneficiaries, or tribal women, health care providers, community leaders, and local governance bodies. All these actors play key roles in determining the effectiveness of health care schemes and in identifying barriers to health care provision. The barrier between the community and government healthcare workers are one of the challenges facing the healthcare schemes for tribal women. It becomes all the more important to understand the challenges at the level of the stakeholders who are providing healthcare schemes to tribal women in the interest of improving health outcomes. Analysing the challenges from the perspectives of the various different people involved the objective of this study

is to understand the challenges and ways to overcome them from different Stakeholder's perspectives in the provision of tribal women's healthcare schemes.

This study fills the gap by providing a comprehensive multi-stakeholder qualitative assessment of reproductive and adolescent healthcare schemes in tribal Jharkhand, mapping challenges and enablers across key health system domains including workforce accountability, service delivery, financing, infrastructure, leadership, and community engagement.

## Materials and methods

An Institutional Ethics Committee (IEC), approval was taken from Manipal Tata Medical College, Jamshedpur, Jharkhand. IEC No: MTMC/IEC/2023/53 dated 16th October, 2023. This was a non-invasive with minimal risk study. The researcher RR and JB went to the assigned community settings with permission from the Medical Officer In-charge (MOIC).

In-depth interviews were conducted by RR and JB. Whereas transcripts preparation and the analysis by derivation of codes and themes were conducted by all the authors (RR, JB, MP, & SIA). RR a male Ph.D. research scholar with experience of total three years in conducting in-depth interviews and focus group discussions, also completed a four-credit course on the qualitative research methods conducted by Prasanna School of Public Health, Manipal Academy of Higher Education, Manipal, India. Whereas JB, and MP female professors and SIA a male professor cum head at the department of community medicine have the expertise in qualitative research methods.

All the researchers established a relationship with participants by visiting the village and obtaining permission at the grassroots level from the gatekeepers such as village heads, healthcare workers, leader from the tribal community, and medical officers. Study participants were informed about the study, its benefits, its role in a doctoral project, and the assurance of confidentiality. No bias was reported in the interviewer's conduct.

This study was based on the qualitative descriptive design using the purposive sampling technique. A face-to-face, in-depth interviews were conducted over a time period of six months, from 2nd December, 2024–2nd June, 2025 in the two blocks from East Singhbhum district, from one of the Indian state Jharkhand. All the in-depth interviews were done at participants home or in the working places such as community hospitals, and village head offices, without any presence of non-participants beside them. Study population included the stakeholders from the receiver's side such as tribal women in the reproductive age (15–49 years), adolescent girl (10–19 years). From the provider's side stakeholders were Accredited Social Health Activists (ASHA), Auxiliary Nurse Midwives (ANM), Sahhiya Sathi (ASHA-Supervisor), Aanganwadi Workers (AWW), Aanganwadi-helper, Medical Officers (MO), Gram Pradhan/Village head (GP), and Gram Mukhiya/Panchayat head (GM). The reproductive age women and adolescent girls were from the tribal communities such as Santhal, Ho, Munda, and Kharia tribes. The reporting of this qualitative study followed the Consolidated Criteria for Reporting Qualitative Research (COREQ) checklist, which has been provided as a Supplementary file S1 Table, COREQ checklist. The protocol of this study is published in the same PlosOne journal [8].

## Data collection

A semi-structured interview guide was designed by RR and JB based on the study objectives and a review of relevant literature. Any discrepancies, ambiguities, or conceptual doubts in the guide were discussed and resolved through consultation with SIA and MP to ensure the contextual relevance. Prior to the main data collection, the interview guide was pilot tested in the field among ten participants, including healthcare workers, tribal women of reproductive age, and adolescent girls, who were not included in the final analysis. The pilot interviews were conducted in settings similar to the actual study sites to assess the clarity, sequencing, cultural appropriateness, and comprehensibility of the questions. Feedback from the pilot testing was used to refine question wording, simplify language, and adjust probing techniques, after which the final version of the interview guide was used for data collection.

Interviewer's guide covered the ice breaking questions as an opening question, Body of the interviewer's guide were the questions related to the key strengths of the current healthcare schemes, government commitments, trainings, and capacity buildings. Major weaknesses such as Lack of infrastructure or accessibility issues hinderance in the implementation of these healthcare programs. Opportunities such as community-based empowerments, and challenges faced in the provision of healthcare schemes, followed by the closing questions suggestions or feedback if any.

A total of eight reproductive healthcare schemes were included in the study. These healthcare schemes were Rashtriya Kishori Swasthya Karyakram (RKSK), POSHAN Abhiyan, Janani Suraksha Yojana (JSY), Janani Sishu Suraksha Karyakaram (JSSK), Anaemia Mukt Bharat (AMB), Pradhan Mantri Matri Vandana Yojana (PMMVY), Pradhan Mantri Surakshit Matritva Abhiyan (PMSMA), and Family Planning Schemes under National Health Mission (NHM) including: ASHA Scheme and Enhanced Compensatory Scheme (ECS).

In-depth interviews were done through audio recordings in Hindi language followed by the field notes. A Participatory Information Sheet (PIS) were explained to the participants. Written Consent with signatures were taken through Informed Consent (IC) forms, in case of minors below the age 18 years assent were taken from their parents/guardians in written. For illiterate participants thumb impression were taken with the signatures of witness in the Informed consent form. In depth interviews were conducted from the twelve villages, under two blocks Potka and Golmuri cum Jugsalai blocks from the East Singhbhum district in Jharkhand, India. Interviews were concluded upon achieving data saturation. This was assessed using a group-wise approach across stakeholder categories, as well as overall thematic saturation. Interviews were continued within each stakeholder group until no new codes or themes emerged, and cross-group comparisons were used to confirm saturation at the overall study level.

Each in-depth interview lasted 10–15 minutes. Although in-depth interviews were conducted using a focused interview guide, the duration of interviews was relatively short. This was primarily due to field-level challenges, including limited participant cooperation in some tribal settings, where respondents tended to provide brief or minimal responses, and time constraints among healthcare providers who were engaged in routine service delivery. These contextual factors may have restricted the depth of probing for certain themes, and future studies may benefit from longer interviews, repeat interactions, or alternative participatory methods to facilitate richer data collection in similar settings.

Transcripts, were developed by translating the Hindi verbatims into English. The developed transcripts were rechecked with the translators inform of language validation. These translators were the school teachers from a private school with expertise in Hindi and English subjects. Also, some transcripts were also returned to the study participants such as medical officers and ANMs, who are literate and well versed with English readings to check the accuracy of the transcripts.

## Data analysis

Data were analysed using thematic analysis following a mixed inductive-deductive approach. Initial codes were generated inductively from the interview transcripts to allow themes to emerge directly from the data. Subsequently, deductive elements were guided by predefined health system domains relevant to reproductive and adolescent health schemes for category section.

All the real names, age, and designation of study participants were coded to maintain the anonymity of the participants. Coding was conducted independently by multiple researchers (RR and JB) using Atlas.ti (version 8). The preliminary codebook was then reviewed and refined through iterative discussions with SIA and MP to resolve discrepancies, enhance conceptual clarity, and ensure consistency across transcripts. Detailed code book is available in the Supplementary file S2 Table, Codebook of the themes and sub themes developed. Final themes were agreed upon through consensus among all authors. Categories such as healthcare Scheme utilization challenges, healthcare Information, healthcare financing, collaborations, healthcare service delivery, healthcare Infrastructure, healthcare accessibility, healthcare workforce, leadership and governance were framed. Based on the categories, themes were developed. Some themes reported were also had cross cutting characteristics which were common under different categories framed.

## Results

There were total 38 study participants interviewed till the data saturation. Data saturation was measured based on the repetition of the same quotes from the different study participants. Characteristics of the study participants including types of stakeholders and the interviews reported till the data saturation are mentioned in the Table 1: Different stakeholder's interviews till the data saturation.

The detailed verbatims of the themes developed are mentioned in the Supplementary file S3 Table, Verbatim's reported from the in-depth interviews. All the themes developed have been discussed under the broad headings of the nine framed categories. These are mentioned as follows.

1. Challenges fac\ed in the provision of reproductive healthcare schemes:

The themes derived under this category are presented in Fig 1.

Fig 1 illustrates the thematic framework of key challenges in providing reproductive healthcare schemes, highlighting system, socio-cultural, infrastructural, workforce, governance, and beneficiary-level barriers identified through qualitative analysis.Providing reproductive healthcare schemes such as JSY, JSSK, PMMVY, PMSMA, AMB, Poshan Abhiyan, RKSK, and ECS, in tribal areas faces numerous challenges across thirteen themes. Key issues include frequent absenteeism of healthcare workers, such as ASHA workers and doctors, who are often only available during emergencies,

Table 1. Different stakeholder's interviews till the data saturation.

| S. No. | Stakeholders | No of Study Participants | Total interview reported till the data saturation (n) |
|---|---|---|---|
| 1. | Village Heads including Gram Panchayat and Gram Pradhan | 4 | 4 |
| 2. | Medical Officers | 4 | 4 |
| 3. | Accredited Social Health Activists (ASHA) and their supervisor | 6 | 6 |
| 4. | Auxiliary Nurse Midwifery (ANM) | 5 | 5 |
| 5. | Aanganwadi Workers (AWW) & their helpers | 5 | 5 |
| 6. | Tribal women in the reproductive age (15–49 years) | 8 | 8 |
| 7. | Tribal adolescent girls (10–19 years) | 6 | 6 |

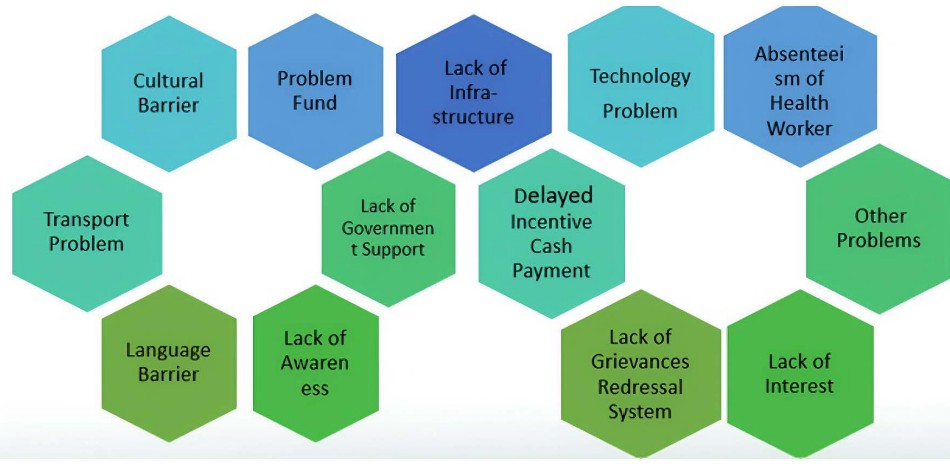

**Fig 1. Themes developed under the category "Challenges faced in the provision of reproductive healthcare schemes.**

leaving routine care neglected. Cultural barriers persist, with a preference for traditional home deliveries assisted by Dai Ma due to religious beliefs like Sarna Dharam. Traditional medicine is favoured over allopathic care due to concerns about side effects. Delayed incentive payments for both beneficiaries and healthcare workers breed frustration and mistrust. Other barriers include low awareness of healthcare schemes, inadequate government support, poor infrastructure, language barriers, funding issues, and difficulties with health technology and transportation.

Quotes:

Respondent: "Language problem is there, we know Hindi but tribal people have their language I am unable to understand sometimes what they are saying." (Female, 32, Medical Officer)

Respondent: "Some of them still prefer local food and medicines. There are some natural herbs we used to take. Because after taking native herbs we will not feel any discomfort." (Female, 33, Tribal Women)

## 2. Collaborations

The category collaborations emphasize on the emergence of two themes, i.e., local leader support and non-governmental organisations collaborations. Leaders such as the Mukhiya play a crucial role in community awareness, though formal healthcare inspections are often missing. NGOs, such as the N Plus Foundation (coded name), contribute to reproductive and adolescent health. However, inconsistent NGO engagement, especially post-COVID-19, has disrupted healthcare continuity, raising concerns about long-term sustainability.

Quote:

Mukhiya ji and N plus foundation (Coded name) is working with us and help us when we are in need help towards health care."(Female, 25, Health Worker)

## 3. Health Accessibility

Under the category health accessibility, reported themes, such as healthcare schemes, service quality, traditional medicine, technological barriers, and transport challenges. Despite improvements from schemes like JSY, many struggle to access full benefits. Disparities in service quality, like inconsistent medicine supply, hinder access. Delays and poor infrastructure diminish service effectiveness. Also study participants were not fully awared with the names of the schemes. They were aware of the schemes that provide only cash benefits. Furthermore, limited awareness and use of telemedicine and mobile health technologies highlight a significant gap in leveraging technology to enhance healthcare access in tribal regions.

Quote:

Respondent: "Government is doing good. There are schemes such as Janani yojana cash benefit of rs 1400. I am not knowing the full name one is Vandana yojana 5000 cash benefit all these things are there. But still tribal people are not aware of the incentives provided under the scheme." (Female, 35, Tribal Women)

## 4. Health Service Delivery

Healthcare workers, especially ASHAs and AWWs, are crucial in delivering services by supporting pregnant women, distributing supplies, and promoting health schemes. However, their efforts are hampered by irregular outreach and supply delays from CHCs. Though cultural barriers are being tackled, additional support is essential. Cash incentives boost engagement, but delayed payments reduce their impact. A major challenge is inadequate government support, especially

financial aid, forcing workers to use personal funds for supplies. Key themes in health service delivery include lack of government support, IEC-based training, trust-building with tribal communities, tailored IEC methods, support for ASHA/ANM/AWW, and technological barriers.

*Quotes:*

*Respondent: "We used to go to Anganwadi centre it is nearby. Infrastructure is nice." (Female, 26, Tribal Women)*

*Respondent: "I would suggest if you will do any awareness sessions, please provide some banners & posters with more pictures in it. It will help the tribal women to grab the content." (Female, 38, Village Leader)*

5. Healthcare Workforce

Various themes were reported such as, supervision, government support, availability of healthcare workers, ASHA/ANM/AWW support, lack of government support, and absenteeism of healthcare workers. The study revealed a shortage of healthcare workers, particularly ANMs and doctors, leading to inadequate service coverage. Absenteeism and irregular engagement of healthcare professionals further strain the system. Many respondents emphasized the need for more staff to ensure regular healthcare delivery especially in emergencies. Despite some improvements in government support, critical gaps in infrastructure and financial assistance persist, limiting the effectiveness of healthcare programs.

*Quotes: "Now facilities are there from the government side related to schemes, such as iron tablets and all, also Kishori yojana is running." (Female, 24, Tribal Women)*

6. Health Information

Key themes included the need for frequent awareness, awareness strategies, school-based programs, and knowledge of reproductive health schemes. A significant barrier to healthcare delivery was the lack of awareness among tribal communities about existing schemes. While some knew about incentives like those under JSY, many were unaware of the full benefits. Cultural preferences for traditional practices further hindered the adoption of modern healthcare. Visual aids, such as posters and street plays, were seen as effective for raising awareness but were inconsistently used.

Quote:

*"Absence of awareness about the available healthcare schemes can result in poor health outcomes within the tribal women. Awareness will be generated, not at one time, two times or three times. We have to motivate them frequently, then only awareness will generate." (Female, 38, Village leader)*

7. Healthcare Financing

Healthcare financing issues are significant, with respondents highlighting delays in receiving cash incentives from various schemes. Incomplete documentation and payment discrepancies further erode trust. Healthcare workers also experience delays in their incentives, impacting motivation and timely service delivery. Key themes identified include cash incentives, funding issues, delayed payments, and insufficient government support.

Quote:

*"We don't receive our cash incentives on time. Higher authorities say fund problem is there."*

*(Female, 38, Village Leader)*

## 8. Healthcare Infrastructure

The themes identified include infrastructure adequacy, service quality, lack of facilities, and technology challenges for healthcare workers. This section explores their perspectives on infrastructure in tribal areas. While some noted improvements, such as well-equipped Anganwadi centres, others pointed out persistent gaps, like the absence of fans and the need for new buildings. Primary Health Centre's (PHCs) often lack essential utilities like electricity and water, while Community Health Centre's (CHCs), though better equipped, suffer from construction delays. Poor infrastructure continues to hinder healthcare delivery and reduce participation in programs, despite some progress in recent years.

Quote:

*"We used to go to Anganwadi centre it is nearby. Infrastructure is nice. But there is no facility of fans. They give medicines & edibles." (Female, 26, Tribal Women)*

## 9. Leadership and governance

Under the category leadership and governance, it emphasizes the roles of family, government, and local leaders in shaping healthcare practices among tribal women. Key themes included family support, maintaining records, government support, leader support, lack of government support, and supervision. Mothers-in-law can both encourage modern healthcare or reinforce traditional practices. Government support has enhanced infrastructure and healthcare access, yet service delivery remains inconsistent. Local leaders, like the Mukhiya and NGO's, are essential in awareness efforts, though their impact varies. Regular supervision and record inspections are vital, but gaps in accountability and inconsistent oversight by authorities affect healthcare quality and efficiency.

Quote:

*"Medical officer sir, Sahiya Sathi, and block trainer used to come frequently to check and inspect the record that we have maintained in the register." (Female, 21, Health Worker)*

## Discussion

This study offers an in-depth examination of the systemic barriers affecting the delivery of reproductive healthcare schemes in tribal areas, interpreted through the lens of established health systems and healthcare access frameworks, particularly the WHO Health System Building Blocks and the dimensions of access-availability, accessibility, acceptability, and affordability [9]. Framing the findings within these conceptual models enables a shift from descriptive reporting to a more explanatory understanding of how interlinked system failures shape reproductive health outcomes in marginalized tribal population.

### Healthcare workforce and service delivery

One of the most critical issues identified is the lack of healthcare personnel, particularly doctors and ASHA workers, in tribal areas. Another is absenteeism, especially when there is no emergency, creates a significant gap in the regular delivery of healthcare services. This problem is compounded by the lack of mechanisms for monitoring and accountability, which are essential to address absenteeism and ensure that healthcare services are delivered consistently. As noted in another studies, absenteeism in rural healthcare systems directly impacts service delivery and erodes community trust [10–12]. This issue needs to be addressed through the development of robust monitoring systems to hold healthcare workers accountable and ensure that services are provided regularly.

## Cultural and community engagements

Cultural barriers pose a significant challenge to healthcare delivery in tribal areas. For instance, many women prefer home deliveries assisted by traditional birth attendants (Dhai-Maa) rather than institutional deliveries. This preference is rooted in religious and cultural beliefs, particularly those associated with Sarna Dharam, and a general mistrust of allopathic medicines due to concerns about side effects. A similar study emphasize that cultural beliefs surrounding childbirth and traditional medicine are common barriers to healthcare acceptance in tribal regions [13]. Another major issue identified in the study is the delay in incentive payments, both to healthcare beneficiaries and workers. These delays contribute to a sense of frustration and mistrust in the healthcare system.

## Financial incentives and healthcare financing

Timely financial incentives are essential for the success of healthcare programs, as they motivate both healthcare workers and community members to actively participate in healthcare initiatives. One of the study researchers stressed the importance of timely and adequate financial incentives in encouraging participation and ensuring the success of health programs [14]. Significant progress has been made in improving healthcare access, with many respondents aware of schemes like JSY and utilizing cash benefits that boosted demand for healthcare services. Similarly, some studies highlighted that the MAMATA scheme not only stimulated demand but also empowered beneficiaries, promoting a sense of independence. Additionally, disparities in healthcare quality, particularly the inconsistent supply of medicines, hinder access to healthcare services [15,16].

## Infrastructure and health information

Lack of government support, inadequate grievance redressal systems, and poor infrastructure further exacerbate the challenges faced in delivering healthcare services. In particular, the study found that many Anganwadi centres in tribal areas are poorly equipped, limiting healthcare access for women and adolescents. These structural and logistical issues must be addressed to enhance the delivery of healthcare services. Studies also mentioned the problems confronted in tribal areas are not only shortfall of medical staff but inadequate number of health centres and its poor accessibility highlights the success of health schemes in remote areas depends heavily on the availability of robust infrastructure. Collaborations with local leaders and NGOs are key enablers in healthcare delivery in tribal areas. Village head and other local leaders also plays a crucial role in raising awareness about health schemes & programs [17,18].

Our study also highlights a lack of awareness and implementation of telemedicine and mobile health technologies, which could potentially bridge geographical barriers. However, the success of these technologies depends on effective training, available infrastructure, and community acceptance. One of the study, emphasize that harnessing the community-based digital solutions driven by developments in science and technology, or digital health, may be the critical next step to facilitate equitable access to education and health care among indigenous people [19].

The scarcity of healthcare workers, particularly ANMs and doctors, is another major challenge in tribal regions. This scarcity, combined with absenteeism, reduces the effectiveness of healthcare schemes and limits service coverage. Despite some improvements in government support, significant gaps remain in terms of financial assistance and infrastructural development. Some studies, argue that healthcare delivery in remote areas will continue to be inadequate without sufficient human resources. There is a need for sufficient doctors and nurses in government health facilities were overwhelmed by high demand due to a shortage of human resources in the tribal areas [6,18].

## Leadership governance and policy implementation

Gaps in government support, weak grievance redressal mechanisms, utilization of maternal and new born services with poor policy implementation further exacerbate challenges which was reported in this study. Other studies also discussed

about the disparities in the utilization of maternal healthcare needs which hampers the policy implementation [20,21]. Similarly, another study done in Odisha also found that, the utilization of maternity and newborn care services was low within the community, with a significant gap between the guidelines outlined by programs like Integrated Management of Neonatal and Childhood Illness (IMNCI) and their actual implementation [22].

Effective leadership and governance are essential to ensure coordinated service delivery, adequate staffing, and infrastructure development [6,16]. Without strengthening these elements, health programs in remote tribal areas may continue to fall short of intended outcomes.

Overall, the study underscores that challenges in tribal healthcare are multifactorial and interconnected across health system domains. Addressing workforce shortages, absenteeism, cultural barriers, financial disincentives, infrastructure deficits, and governance gaps is essential for improving reproductive health outcomes. Applying a health systems framework allows these findings to be understood not merely as isolated observations but as systemic issues requiring integrated and multi-level intervention.

### Limitations of the study

The study has several limitations. Although in-depth interviews were conducted using a focused guide, the duration was relatively short (10–15 minutes) due to field constraints. Participants, particularly from tribal communities, often provided brief responses, and healthcare workers faced time limitations, which may have restricted the depth of information obtained. Additionally, the findings may be influenced by subjective bias inherent in participants' responses and the researchers' interpretations. Finally, the results are context-specific and may not be generalizable to broader populations beyond the study setting.

### Recommendation

- Strengthen workforce accountability: by addressing absenteeism through regular supervision, incentives, and community-based monitoring systems.

- Enhance infrastructure with urgent construction and renovation of Anganwadi Center's and PHCs, along with ensuring water, electricity, and transport facilities.

- Ensure timely disbursement of incentives to both beneficiaries and healthcare workers to maintain trust and motivation.

- Promote culturally sensitive IEC strategies through visual tools, street plays, and tribal language-based materials tailored to local needs.

- Build trust through community leaders and NGOs by formalizing partnerships to sustain awareness and outreach efforts.

- Implement continuous training for ASHA, ANM, and AWW on digital tools and reproductive health schemes.

- Address language and cultural barriers via local translators and by integrating traditional beliefs with modern healthcare promotion.

### Conclusion

The study reveals persistent challenges in the awareness and utilization of reproductive health schemes among tribal populations. From the provider side, healthcare workers such as ASHAs, ANMs, AWWs, medical officers, and village heads face major barriers including staff absenteeism, weak government support, and inadequate infrastructure. Nevertheless, supervision by higher authorities and the delivery of quality services were recognized as positive factors that can be strengthened to improve service provision. From the receiver side, adolescent girls and women of reproductive age exhibit

low levels of reproductive health knowledge, with cultural barriers, reliance on traditional medicine, financial difficulties, transport challenges, and limited interest in scheme utilization acting as key obstacles. Importantly, stakeholders emphasized the urgent need for frequent, school-based awareness campaigns, community-oriented programs, and culturally sensitive IEC strategies to bridge these gaps. Addressing both provider- and receiver-side challenges through stronger governance, better resource allocation, and sustained awareness interventions will be critical to improving reproductive health outcomes in tribal areas.

## Supporting information

**S1 Table. COREQ checklist.**
(PDF)

**S2 Table. Codebook of the themes and sub themes developed.**
(DOCX)

**S3 Table. Verbatim's reported from the in-depth interviews.**
(DOCX)

## Author contributions

**Conceptualization:** Rohit Raj, Jarina Begum, Syed Irfan Ali, Manasee Panda.

**Data curation:** Rohit Raj, Jarina Begum.

**Formal analysis:** Rohit Raj, Jarina Begum, Syed Irfan Ali, Manasee Panda.

**Investigation:** Rohit Raj, Jarina Begum, Syed Irfan Ali, Manasee Panda.

**Methodology:** Rohit Raj, Jarina Begum, Syed Irfan Ali, Manasee Panda.

**Project administration:** Rohit Raj, Jarina Begum, Syed Irfan Ali, Manasee Panda.

**Resources:** Rohit Raj, Jarina Begum, Syed Irfan Ali, Manasee Panda.

**Software:** Rohit Raj, Jarina Begum.

**Supervision:** Rohit Raj, Jarina Begum, Syed Irfan Ali, Manasee Panda.

**Validation:** Rohit Raj, Jarina Begum, Syed Irfan Ali, Manasee Panda.

**Visualization:** Rohit Raj, Jarina Begum, Syed Irfan Ali, Manasee Panda.

**Writing – original draft:** Rohit Raj, Jarina Begum.

**Writing – review & editing:** Rohit Raj, Jarina Begum, Syed Irfan Ali, Manasee Panda.

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
