## [Decision Letter · Decision Letter 0]

13 Jan 2026

Dear Dr. Begum,

Thank you for submitting your manuscript to PLOS ONE. After careful consideration, we feel that it has merit but does not fully meet PLOS ONE’s publication criteria as it currently stands. Therefore, we invite you to submit a revised version of the manuscript that addresses the points raised during the review process.

Kindly address the comments provided by reviewer 2.

We look forward to receiving your revised manuscript.

Kind regards,

Yogesh Kumar Jain, PhD

Academic Editor

PLOS One

Journal Requirements:

[The authors have declared that no competing interests exist.].

We note that one or more of the authors are employed by a commercial company: Manipal Tata Medical College.

4. We note that this data set consists of interview transcripts. Can you please confirm that all participants gave consent for interview transcript to be published?

If they DID provide consent for these transcripts to be published, please also confirm that the transcripts do not contain any potentially identifying information (or let us know if the participants consented to having their personal details published and made publicly available). We consider the following details to be identifying information:

- Names, nicknames, and initials

- Age more specific than round numbers

- GPS coordinates, physical addresses, IP addresses, email addresses

- Information in small sample sizes (e.g. 40 students from X class in X year at X university)

- Specific dates (e.g. visit dates, interview dates)

- ID numbers

Or, if the participants DID NOT provide consent for these transcripts to be published:

- Provide a de-identified version of the data or excerpts of interview responses

- Provide information regarding how these transcripts can be accessed by researchers who meet the criteria for access to confidential data, including:

a) the grounds for restriction

b) the name of the ethics committee, Institutional Review Board, or third-party organization that is imposing sharing restrictions on the data

c) a non-author, institutional point of contact that is able to field data access queries, in the interest of maintaining long-term data accessibility.

d) Any relevant data set names, URLs, DOIs, etc. that an independent researcher would need in order to request your minimal data set.

For further information on sharing data that contains sensitive participant information, please see: https://journals.plos.org/plosone/s/data-availability#loc-human-research-participant-data-and-other-sensitive-data

If there are ethical, legal, or third-party restrictions upon your dataset, you must provide all of the following details (https://journals.plos.org/plosone/s/data-availability#loc-acceptable-data-access-restrictions):

1. A complete description of the dataset

2. The nature of the restrictions upon the data (ethical, legal, or owned by a third party) and the reasoning behind them

3. The full name of the body imposing the restrictions upon your dataset (ethics committee, institution, data access committee, etc.)

4. If the data are owned by a third party, confirmation of whether the authors received any special privileges in accessing the data that other researchers would not have

5. Direct, non-author contact information (preferably email) for the body imposing the restrictions upon the data, to which data access requests can be sent

Reviewers' comments:

Reviewer's Responses to Questions

**Comments to the Author**

1. Is the manuscript technically sound, and do the data support the conclusions?

Reviewer #1: Yes

Reviewer #2: Partly

2. Has the statistical analysis been performed appropriately and rigorously?

Reviewer #1: Yes

Reviewer #2: Yes

3. Have the authors made all data underlying the findings in their manuscript fully available?

Reviewer #1: Yes

Reviewer #2: Yes

4. Is the manuscript presented in an intelligible fashion and written in standard English?

Reviewer #1: Yes

Reviewer #2: No

Reviewer #1: The study is a well planned qualitative study addressing a crucial region specific lacuna. The leading questions and interviews have given answers to the questions raised .The study results can be noted and followed up by the stakeholders for addressing the lacuna

Reviewer #2: Regarding the contribution of the study, the added value compared to previous studies in Indian populations is not clearly seen. It is suggested to include 1 or 2 paragraphs on which specific gap this study fills. This is in order for the reader to understand why the results matter for policies and programs.

Regarding the design, the analytical approach is not specified. It is suggested to state “qualitative descriptive design”, or “interpretive descriptive approach”, or “case study design”, or whatever was actually done. The use of a standard reference such as COREQ as a checklist should be added, and it should also be mentioned who designed the interview guides and how the pilot was carried out.

Regarding saturation, the process should be detailed, clarifying whether saturation was sought by group or in a general/global way. This is important to provide credibility to the study.

Regarding the depth of the interviews, a duration of 10 to 15 minutes is reported. It is suggested to be more modest and use “semi-structured interviews” unless the guides were very focused and field notes were collected. Otherwise, this should be included as a limitation.

Regarding the qualitative analysis procedure, it is not specified whether it was inductive, deductive, or mixed, nor the number of coders.

Regarding the discussion, it should be developed using relevant conceptual frameworks such as the health system, accessibility, etc. The same findings are repeated many times but with little theoretical depth.

In the ethics section, there are elements written in the future tense; the description should be unified in the past tense

**Do you want your identity to be public for this peer review?** For information about this choice, including consent withdrawal, please see our Privacy Policy

Reviewer #1: No

Reviewer #2: No

---

## [Author Response · Author response to Decision Letter 1]

15 Jan 2026

Reviewer #1:

The study is a well-planned qualitative study addressing a crucial region-specific lacuna. The leading questions and interviews have given answers to the questions raised. The study results can be noted and followed up by the stakeholders for addressing the lacunae.

Answer: We sincerely thank Reviewer 1 for their positive and encouraging feedback, and we appreciate the recognition of the study’s methodological rigor, region-specific relevance, and potential utility for stakeholders in addressing identified lacunae.

Reviewer #2:

1. Regarding the contribution of the study, the added value compared to previous studies in Indian populations is not clearly seen. It is suggested to include 1 or 2 paragraphs on which specific gap this study fills. This is in order for the reader to understand why the results matter for policies and programs.

• We thank the reviewer for this important suggestion. To clearly articulate the study’s added value, we have revised the Introduction sections by adding two dedicated paragraphs that explicitly highlight the specific gap addressed by this study.

2. Regarding the design, the analytical approach is not specified. It is suggested to state “qualitative descriptive design”, or “interpretive descriptive approach”, or “case study design”, or whatever was actually done. The use of a standard reference such as COREQ as a checklist should be added, and it should also be mentioned who designed the interview guides and how the pilot was carried out.

• We thank the reviewer for this constructive suggestion. We have now clearly specified the study as a qualitative descriptive design in the Materials and Methods section to enhance methodological clarity.

• Additionally, we have stated that the study reporting follows the Consolidated Criteria for Reporting Qualitative Research (COREQ) checklist, which has been added to the Methods section and included as a supplementary file.

• We have also clarified that the semi-structured interview guides were self-developed by the research team, informed by existing literature and study objectives, and that these guides were pilot tested among ten stakeholders to assess clarity, relevance, and cultural appropriateness before final data collection.

3. Regarding saturation, the process should be detailed, clarifying whether saturation was sought by group or in a general/global way. This is important to provide credibility to the study.

• We thank the reviewer for highlighting this important methodological aspect. We have now elaborated on the data saturation process in the Methods section, clarifying that saturation was assessed using a group-wise approach across stakeholder categories, as well as overall thematic saturation. Interviews were continued within each stakeholder group until no new codes or themes emerged, and cross-group comparisons were used to confirm saturation at the overall study level. These revisions have been incorporated to enhance transparency and methodological rigor.

4. Regarding the depth of the interviews, a duration of 10 to 15 minutes is reported. It is suggested to be more modest and use “semi-structured interviews” unless the guides were very focused and field notes were collected. Otherwise, this should be included as a limitation.

• We thank the reviewer for this important clarification. The study employed in-depth interviews using a focused interview guide; however, we acknowledge that the interview duration was relatively short (10–15 minutes) due to contextual field constraints. Many participants, particularly from tribal communities, provided brief or one-word responses, and several healthcare workers reported time constraints related to service delivery responsibilities, which limited prolonged engagement. We have now explicitly acknowledged this as a study limitation in the manuscript to ensure transparency and appropriate interpretation of the depth of findings.

5. Regarding the qualitative analysis procedure, it is not specified whether it was inductive, deductive, or mixed, nor the number of coders.

• We thank the reviewer for this important methodological clarification. We have now specified that the thematic analysis followed a mixed inductive-deductive approach. Initial coding was conducted inductively from the data, while deductive elements were guided by predefined health system domains relevant to reproductive and adolescent health schemes for category section. We have also clarified that multiple coders were involved in the analysis, with independent coding followed by team discussions to resolve discrepancies and refine the codebook. These details have been added to the Data analysis section to enhance transparency and rigor.

6. Regarding the discussion, it should be developed using relevant conceptual frameworks such as the health system, accessibility, etc. The same findings are repeated many times but with little theoretical depth.

• We have updated the discussion under the sub headings based on WHO building blocks of health system. To add the evidence we have tried to incorporate more depth with references to the discussion part.

7. In the ethics section, there are elements written in the future tense; the description should be unified in the past tense.

• Corrected.

---

## [Decision Letter · Decision Letter 1]

11 Feb 2026

Multi-stakeholder perspectives on reproductive and adolescent healthcare schemes in tribal regions of India: A qualitative study.

PONE-D-25-44960R1

Dear Dr. Begum,

We’re pleased to inform you that your manuscript has been judged scientifically suitable for publication and will be formally accepted for publication once it meets all outstanding technical requirements.

Kind regards,

Yogesh Kumar Jain, PhD

Academic Editor

PLOS One

Additional Editor Comments (optional):

Reviewers' comments:

Reviewer's Responses to Questions

**Comments to the Author**

Reviewer #2: All comments have been addressed

2. Is the manuscript technically sound, and do the data support the conclusions?

Reviewer #2: Yes

3. Has the statistical analysis been performed appropriately and rigorously?

Reviewer #2: Yes

4. Have the authors made all data underlying the findings in their manuscript fully available?

Reviewer #2: Yes

5. Is the manuscript presented in an intelligible fashion and written in standard English?

Reviewer #2: Yes

Reviewer #2: I have reviewed the R1 version of the manuscript and can confirm that it addresses all previously noted observations, responding to each point appropriately and comprehensively. I would also like to highlight that the manuscript demonstrates additional improvements, particularly in the restructured discussion using relevant conceptual frameworks and in the clearer explanation of the analytical approach, which substantially strengthens both methodological rigor and overall coherence.

**Do you want your identity to be public for this peer review?** For information about this choice, including consent withdrawal, please see our Privacy Policy

Reviewer #2: No

---

## [Editor Report · Acceptance letter]

PONE-D-25-44960R1

PLOS One

Dear Dr. Begum,

I'm pleased to inform you that your manuscript has been deemed suitable for publication in PLOS One. Congratulations! Your manuscript is now being handed over to our production team.

Kind regards,

on behalf of

Dr. Yogesh Kumar Jain

Academic Editor

PLOS One